**Brief Investigation**

# High levels of intra-strain structural variation in *Drosophila simulans* X pericentric heterochromatin

Cécile Courret [ID],* Amanda M. Larracuente [ID] *

Department of Biology, University of Rochester, Rochester, NY 14627, USA

*Corresponding author: Department of Biology, University of Rochester, Rochester, NY 14627, USA. Email: cecile.courret@rochester.edu; *Corresponding author: Department of Biology, University of Rochester, Rochester, NY 14627, USA. Email: alarracu@bio.rochester.edu

Large genome structural variations can impact genome regulation and integrity. Repeat-rich regions like pericentric heterochromatin are vulnerable to structural rearrangements although we know little about how often these rearrangements occur over evolutionary time. Repetitive genome regions are particularly difficult to study with genomic approaches, as they are missing from most genome assemblies. However, cytogenetic approaches offer a direct way to detect large rearrangements involving pericentric heterochromatin. Here, we use a cytogenetic approach to reveal large structural rearrangements associated with the X pericentromeric region of *Drosophila simulans*. These rearrangements involve large blocks of satellite DNA—the *500-bp* and *Rsp-like* satellites—which colocalize in the X pericentromeric heterochromatin. We find that this region is polymorphic not only among different strains, but between isolates of the same strain from different labs, and even within individual isolates. On the one hand, our observations raise questions regarding the potential impact of such variation at the phenotypic level and our ability to control for such genetic variability. On the other hand, this highlights the very rapid turnover of the pericentric heterochromatin most likely associated with genomic instability of the X pericentromere. It represents a unique opportunity to study the dynamics of pericentric heterochromatin, the evolution of associated satellites on a very short time scale, and to better understand how structural variation arises.

Keywords: structural variation; pericentromeric heterochromatin; satellite repeats; Drosophila

## Introduction

Structural variants are duplicated, deleted, transposed, or inverted sequences, that can contribute to complex traits (Sudmant et al. 2015; Chakraborty et al. 2019), diseases (Stankiewicz and Lupski 2010), and genome evolution (Chakraborty et al. 2021). Variants involving rearrangements of large genome regions, such as chromosomal translocations and inversions, are associated with diseases involving intellectual disabilities and cancers (Weischenfeldt et al. 2013). Pericentric heterochromatin is rich in repetitive sequences like transposable elements and satellite DNAs (Charlesworth et al. 1986) and may be particularly prone to structural rearrangements from replication stress, nonhomologous recombination, transposable element activity, and a decreased efficiency of some DNA repair pathways (reviewed in Janssen et al. 2018). Structural rearrangements in pericentric heterochromatin may have consequences: although the density of conventional protein-coding genes is low, these regions have roles in genome defense (Andersen et al. 2017), coordinating chromosome segregation and nuclear organization (Folco et al. 2008; Peng and Karpen 2009), and genomic stability (Janssen et al. 2018).

Repeats in the pericentric heterochromatin are highly dynamic over long evolutionary time periods (Lohe and Roberts 1988), as species tend to have their own unique profiles of pericentric repeats. This divergence in the pericentric heterochromatin can lead to genetic incompatibilities between closely related species (Ferree and Barbash 2009; Cattani et al. 2012; Jagannathan et al. 2017; Jagannathan and Yamashita 2021). We know less about the dynamics of pericentric heterochromatin and its functional consequences over short evolutionary timescales, although satellite DNA copy number varies within species (e.g. Wei et al. 2014) and can be associated with chromosome rearrangements (Flynn et al. 2023). However, some functional variation within species maps to highly heterochromatic regions of the genome. For example, variation in Y-linked heterochromatin can impact gene expression across the genome and affect male fertility (Dimitri and Pisano 1989; Chippindale and Rice 2001; Lemos et al. 2008; Sackton et al. 2011; Brown et al. 2020).

The repetitive nature of pericentric heterochromatin makes it difficult to study at the genomic level (Treangen and Salzberg 2012), although the relatively compact genomes of Drosophila species make them mighty models for repeat biology. Drosophila species have a large genetic toolkit and many Drosophila species can be isogenized and inbred, making the genome homozygous and amenable to experiments (Hoskins et al. 2015; Hales et al. 2015). High quality genome assemblies exist for species of the *melanogaster* clade [e.g. *Drosophila melanogaster* (Chang and Larracuente 2019), *Drosophila simulans*, *Drosophila mauritiana*, and *Drosophila sechellia* (Chakraborty et al. 2021; Chang et al. 2022)]. Comparing these assemblies revealed structural divergence between species that may contribute to

important phenotypes. Structural rearrangements involving pericentric heterochromatin are difficult to ascertain with genomic approaches—the most densely repetitive regions of the genome including large blocks of tandem satellite repeats are not yet fully assembled (Chakraborty et al. 2021; Chang et al. 2022). However, cytogenetic approaches indicate that the distribution and type of heterochromatic satellite repeats differ even between these closely related species (Larracuente 2014; Jagannathan et al. 2017; Sproul et al. 2020), implying that large structural variations in repetitive regions contribute to species divergence. Large structural rearrangements in pericentromeric satellite repeats *within* species are less well documented.

Here, we describe striking structural variation in the pericentric heterochromatin of the *X* chromosome in *D. simulans*. We use a cytogenetic approach to document high levels of structural polymorphism in satellite DNAs in the X pericentromere: *Rsp-like* and 500-*bp* satellites. *Rsp-like* is a complex satellite specific to the X pericentromere in *D. simulans* (Sproul et al. 2020) and the 500-*bp* satellite is associated with the centromere and pericentromere of the *X* chromosome and the autosomes in *D. simulans* (Talbert et al. 2018; Courret et al. 2023a). The structural polymorphisms we detect involve large blocks of satellite repeats and occur between different strains, within a strain, and even within individual isolates of strains kept in a single lab. This extreme structural polymorphism may not be conspicuous at the DNA sequencing level, but affects large regions of the pericentromere, and could conceivably have functional impacts.

## Materials and methods
### Fly strains
We use "strain" to refer to a genotype and give a unique name to "isolates", which are lineages of a strain from a particular lab. We have 3 isolates of the $w^{501}$ strains that originated from 3 different laboratories as follows: Larracuente ($w501-i1$), Presgraves ($w501-i2$), and Andolfatto ($w501-i3$). $w501-i1$ and $w501-i2$ have a common origin, but have been maintained separately for 7 years. We have 2 isolates of the $w^{XD1}$ strain that originated from 2 different labs as follows: Presgraves ($wXD1-i1$) and Meiklejohn ($wXD1-i2$). The $wXD1-i2$ isolate originated from the $wXD1-i1$ isolate ~10 years ago. We also used other non-white isofemale *D. simulans* strains as follows: SR (collected from *Seychelles* in 1981), ST8 (collected from *Tunisia* in 1983), C167.4 (collected from *Kenya* in 1973), sim006 (collected from *California* in 1961) (described in Courret et al. 2023b).

### Fluorescence in situ hybridization
We performed FISH using primary oligopaint probes for *Rsp-like* and 500-*bp* (Courret et al. 2023a) coupled with sec6 and sec5 adaptors (Beliveau et al. 2014). Sec5 is coupled with Cy5 while sec6 is coupled with Cy3. We dissected brains from third instar larvae in PBS, incubated 8 min in 0.5% sodium citrate. We fixed for 6 min in 4% formaldehyde, 45% acetic acid before squashing. We squashed the brains between the slide and coverslip and before immersing in liquid nitrogen. After 10 min in 100% ethanol, we air dried slides for at least 1 hour before proceeding to the hybridization. For the hybridization, we used 20 pmol of primary probes and 80 pmol of the secondary probes in 50 µL of hybridization buffer (50% formamide, 10% dextran sulfate, 2xSSC). We heated slides for 5 min at 95°C to denature and incubated them overnight at 37°C in a humid chamber. We then washed the slides 3 times for 5 min with 4XSSCT and 3 times for 5 min with 0.1SSC before mounting in slowfade DAPI. We imaged using a LEICA DM5500 microscope and cropped and pseudocolored the images using Fiji.

**Table 1.** A summary of structural variation involving the 500-*bp* and *Rsp-like* satellites in the *X* chromosome pericentric heterochromatin within and between isolates of *D. simulans* strains.

| Strain (isolate) | No. of individuals | X chromosome frequencies | | |
| --- | --- | --- | --- | --- |
| | | 1-Foci | 2-Foci | 3-Foci |
| $w^{501}$ ($w501-i1$) | 22 | 0 | 0.34 | 0.66 |
| $w^{501}$ ($w501-i3$) | 19 | 0 | 0.93 | 0.07 |
| $w^{501}$ ($w501-i2$) | 19 | 0.21 | 0.79 | 0 |
| $w^{XD1}$ ($wXD1-i2$) | 40 | 0.14 | 0 | 0.86 |
| $w^{XD1}$ ($wXD1-i1$) | 22 | 0 | 0 | 1 |
| SR | 18 | 1 | 0 | 0 |
| C167.4 | 18 | 1 | 0 | 0 |
| ST8 | 20 | 1 | 0 | 0 |
| Sim006 | 20 | 1 | 0 | 0 |

The isolate identities for $w^{501}$ and $w^{XD1}$ are indicated in parentheses. We report the number of individuals (i.e. brains, which includes both males and females) tested: all spreads examined within an individual brain were consistent (see Materials and methods). We report the proportion of 1-, 2-, or 3-focus *X* chromosomes among individuals from each isolate. The detailed genotype of each individual tested is presented in Supplementary Table 1.

We analyzed 4–10 mitotic spreads for each individual brain to determine without ambiguity the number of foci carried by the X chromosomes. We confirmed that all spreads within an individual brain had the same number of foci. To estimate the allele frequency in each isolate, around 20 individual brains were tested, both male and female (full genotype details in Supplementary Table 1). The frequency reported in Table 1 corresponds to the frequency of each type of X chromosome among all individual brains tested.

### Genotyping and genome analysis
We designed primers around SNPs located on the X chromosome. The primer position—alleles on Segkk236 from the reference genome in Chang et al. (2022) and sequences are as follows: 9814904-T/G (forward primer—GCAAAGTCTTTTAAGCGCGC and reverse primer—CCGGGGGAAAATCTGCTTCT); 17904265-A/G (forward primer—GTTGTCGCTCTCCTTGACCA and reverse primer—GCTGGCCATCTTCACCATCT); and 18025547-C/T (forward primer—CTGCTCCGCGTGTATATGGT and reverse primer—ACAGTTCGCGATGAGCTTCT). For each primer pair, we performed a PCR with NEB Taq polymerase (NEB #M0495) following the manufacturer's instructions (hybridization temperature: 53°). We sequenced each PCR product using the Sanger method (ACGT company) and visualized sequence profiles using Geneious.

We downloaded reads for $w^{XD1}$ (SRR8247551; Meiklejohn et al. 2018), ST8, SR, and C167.4 (PRJNA905841; Courret et al. 2023b), and $w^{501}$ (SRR520334; Hu et al. 2013), trimmed and processed reads with trimgalore (v0.6.2) (Krueger et al. 2021) (–paired –nextera –length 75 –phred33 –fastqc). We mapped reads with *BWA-MEM* (v0.17 default parameters) to the *D. simulans* genome assembly (Chang et al. 2022) and estimated coverage (in reads per million) with bamCoverage (-bs 1000) in deeptools (v3.5.1) (Ramírez et al. 2016) across the X chromosome. We plotted in R to look for large-scale differences in coverage that would suggest structural polymorphisms.

To estimate the per-site heterozygosity, we called SNPs using bcftools (v1.6) (Li 2011) *mpileup* and *call* commands, keeping all sites. We filtered the vcf file using vcftools (v0.1.15/b1) (Danecek et al. 2011) (–remove-indels –minQ 30 –minDP 10 –maxDP 200) and then extracted the number of homozygous and heterozygous sites using the bcftools *stats* command.

## Results

We focus our study on 2 commonly used *D. simulans* lab strains as follows: $w^{501}$ and $w^{XD1}$. Both carry a *white* mutation on the X chromosome, conferring the white-eyed phenotype. These inbred strains are frequently used for genetic manipulation (Stern et al. 2017) or genetic mapping (Matute and Ayroles 2014; Meiklejohn et al. 2018) and have abundant genomic resources (Garrigan et al. 2012; Hu et al. 2013; Chakraborty et al. 2021; Chang et al. 2022).

We collected isolates of the $w^{501}$ strain from 3 different laboratories ($w501$-$i1$; $w501$-$i2$, and $w501$-$i3$). $w501$-$i1$ and $w501$-$i2$ have a common origin but have been maintained separately for 7 years (91–119 generations). The $w501$-$i3$ isolate was maintained independently. We also collected isolates of the $w^{XD1}$ strains from 2 different labs as follows: $wXD1$-$i1$ and $wXD1$-$i2$. The $wXD1$-$i2$ originated from the $wXD1$-$i1$ strains 10 years ago (130–170 generations).

The 2 satellites that we use as markers for pericentric structural variation, *500-bp* and *Rsp-like*, are adjacent on the X chromosome and their localization pattern is always similar (i.e. in adjacent blocks). We did not observe any genotypes where *500-bp* and *Rsp-like* did not co-vary in the number of foci. We show that these blocks are highly variable both within and between strains. We observe 3 general colocalization patterns for *500-bp* and *Rsp-like* at 1, 2, or 3 foci in the X pericentric heterochromatin.

### Structural variation within and between isolates of a single strain

The 3 isolates of the $w^{501}$ strain appear to be polymorphic both between and within isolates. The $w501$-$i1$ and $w501$-$i3$ isolates are polymorphic for 2- and 3-focus X chromosomes (Fig. 1a and c). Within the $w501$-$i1$ isolate, we estimated the frequency of the 3-focus and 2-focus X chromosomes at 66 and 34%, respectively (Table 1); while the $w501$-$i3$ has estimated frequencies of 93 and 7%, respectively (Table 1). $w501$-$i2$ shows both 2- and 1-focus X

chromosomes (Fig. 1b), at estimated frequencies of 79 and 21%, respectively (Table 1).

This degree of polymorphism and divergence within a single strain is surprising, as the $w501$-$i1$ isolate originated from the $w501$-$i2$ isolate only 7 years ago (91–119 generations). This suggests that duplication events in the pericentromeric region happened recently and may happen recurrently.

We observe similarly striking structural variation in the pericentromeric region of the $w^{XD1}$ X chromosomes. Consistent with previous observations (Sproul et al. 2020), we find that the $wXD1$-$i1$ X chromosome pericentromere has a 3-focus pattern (Fig. 2a). However, the $wXD1$-$i2$ X chromosome pericentromeric region appears to be polymorphic for the 1-focus and 3-focus patterns (Fig. 2b), with estimated frequencies of 14 and 86%, respectively (Table 1).

Structural polymorphisms involving large blocks of the *Rsp-like* and *500-bp* satellite repeats may generally be detectable through differences in read depth (Larracuente 2014). However, when these polymorphisms exist within a single isolate, they are not obvious in genomic data (Supplementary Fig. 1). In our analysis of sequencing libraries created from pooled individuals, detecting alternative alleles based on read depth is extremely challenging, as it will depend on the frequency of alternative alleles in the pools. Biases in library preparation, tissue, and DNA extraction can all contribute to variation in read mapping in repetitive sequences between biological replicates (Shinde 2003; Aird et al. 2011; Ross et al. 2013; Wei et al. 2018). We suggest that true structural polymorphisms, either between individuals of a single isolate or between tissue and cells within an individual, can also contribute to variable read coverage. We would need multiple biological replicates from the same isolates and, ideally, a contiguous assembly of pericentric heterochromatin to assess the potential for recovering information about these structural rearrangements in genomic data. Currently, a cytogenetic approach is necessary to characterize such structural polymorphisms, especially within isolates.

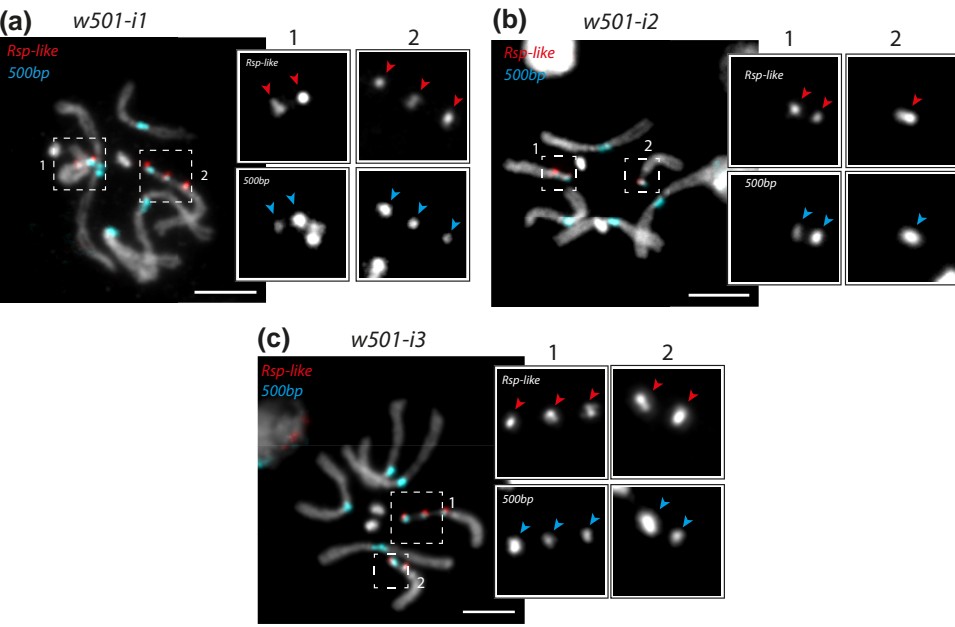

**Fig. 1.** FISH on mitotic chromosomes from larval brain in (a) $w501$-$i1$, (b) $w501$-$i2$, and (c) $w501$-$i3$ strains. We used oligopaint probes targeting the *Rsp-like* (first inset row) and *500-bp* (second inset row) satellites. The scale bar represents 5 μm. The inset zooms in on the X chromosome revealing a heterozygote for a 2-focus (1) and a 3-focus (2) X chromosome in $w501$-$i1$ (a), a heterozygote for a 2-focus (1) and a 1-focus (2) X chromosome in $w501$-$i2$ (b), and a heterozygote for a 3-focus (1) and a 2-focus (2) X chromosome in $w501$-$i3$. The arrows within the inset point to each foci associated with the X chromosome.

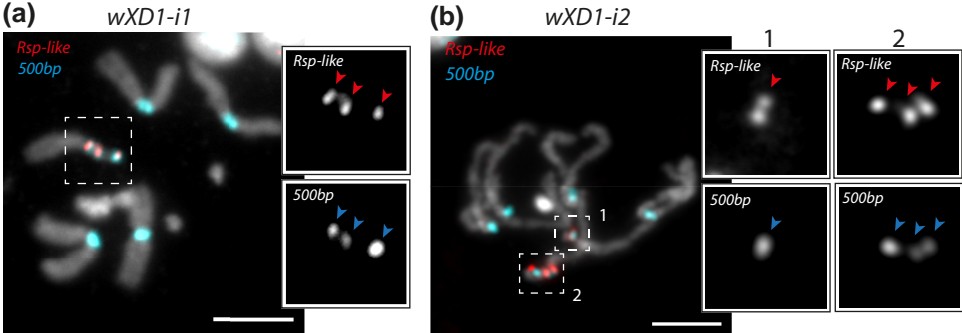

**Fig. 2.** FISH on mitotic chromosomes from larval brains of (a) *wXD1-i1* and (b) *wXD1-i2* strains. We used oligopaint probes targeting the *Rsp-like* (first inset row) and *500-bp* (second inset row) satellites. The scale bar represents 5 μm. The inset zooms in on the X chromosome revealing a 3-focus X chromosome in *wXD1-i1* (a) and a heterozygote for a 1-focus (1) and a 3-focus (2) X chromosome in *wXD1-i2* (b). The arrows within the inset point to each focus associated with the X chromosome.

These white-eyed lab strains have independent origins and therefore, these structural mutations should also be independent. To be sure that the structural variation is not due to strain contamination and/or recombination between the 2 white-eyed lab strains, we genotyped the X chromosomes. We designed primers to genotype 3 SNPs that allow us to differentiate $w^{XD1}$ and $w^{501}$ X chromosomes by PCR re-sequencing. As expected, if pericentromeric variation is due to structural polymorphisms within an X chromosome, the different $w^{501}$ isolates carry the same alleles and the $w^{XD1}$ isolates carry the same alternative alleles at all 3 sites. This suggests that the structural variants arose on their respective X chromosome backgrounds and that the X pericentric heterochromatin is likely unstable in these white-eyed lab strains.

### Within-isolate structural variation seems limited to lab strains

To understand if the chromosomal instability is strain or species specific, we studied satellite organization in 4 different *D. simulans* strains that do not carry *white* mutations as follows: *SR, ST8, sim006,* and *C167.4*. Each of these strains has a single focus of *Rsp-like* and *500-bp* in their X pericentric heterochromatin (Fig. 3). While more strains should be tested in the future, this pattern suggests that the large structural variations may be limited to the $w^{501}$ and $w^{XD1}$ strains.

Isogenization should purge any segregating sequence variants (including structural ones) within strains, although sequence variation may exist due to: (1) mutations that accumulate over time while strains are maintained in labs (Lack et al. 2016); and (2) residual heterozygosity from incomplete inbreeding or linkage to balanced deleterious mutations that cannot be made homozygous. To determine if the structural polymorphism correlates with the extent of inbreeding of each strain, we estimate the per-site heterozygosity (H) of the X chromosome in available genomic data (Hu et al. 2013; Meiklejohn et al. 2018; Courret et al. 2023b). Despite being polymorphic in the X pericentromere, we estimate very low levels of per-site heterozygosity across the X chromosome arm in $wXD1-i2$ ($H = 1.254 \times 10^{-5}$) and $w501-i3$ ($H = 5.93 \times 10^{-5}$). The nonwhite strains appear less inbred—*ST8* ($H = 0.000468$), *SR* ($H = 0.000733$), and *C167.4* ($H = 0.000459$), and similar to a previous estimate for the *sim006* strain ($H = 0.00039$) (Kim et al. 2021).

Therefore, the structural polymorphism is in the strains with the lowest heterozygosity across the X chromosome arm, further supporting our hypothesis that the structural variants arose recently and may be associated with genomic instability in the X pericentromere.

## Discussion

In summary, we find large X-linked structural polymorphisms segregating within single isolates of 2 commonly used lab strains of *D. simulans*. These types of polymorphisms are not obvious in genomic data, although they may contribute to variation in read depth between biological replicates in repetitive regions. Because we observe different variants even within single isolates of the same strain (i.e. within single vials of flies), we hypothesize that this region of the X pericentromere is unstable and associated with recurrent structural rearrangements. We cannot completely rule out the possibility that these variants were already segregating in the original strains and then sorted differently between lab isolates. Labs may differ in their maintenance conditions, which may impose different selection pressures. Different isolates of the same strain maintained in different labs can accumulate isolate-specific TE landscapes (Rahman et al. 2015). Further experiments are necessary to determine the mutation rate in the X pericentromere. A recent origin for these structural variants appears more likely based on multiple observations. First, if there was a pre-existing variation, we would expect more similarity between the $w501-i1$ and $w501-i2$ isolates, based on their recent history, than between $w501-i1$ and $w501-i3$. Second, 2 independent strains ($w^{501}$ and $w^{XD1}$) exhibit structural polymorphism in the same region, suggesting that this X pericentric heterochromatin may experience genomic instability. Finally, the 2 white strains where we see the variation are highly inbred compared to the 4 nonwhite strains which do not have detectable structural polymorphisms.

The structural variation we observe may have functional implications, as pericentric heterochromatin has effects on chromosome dynamics (Dernburg et al. 1996; Karpen et al. 1996), genome stability (Peng and Karpen 2009), genome structure (Falk et al. 2019; Lee et al. 2020), and nuclear organization. These regions also contain, or flank, essential genetic elements, including the centromeres. For example, variation in pericentromeres may affect adjacent centromeres (Kumon et al. 2021; Jagannathan and Yamashita 2021), chromosome structures that are essential for coordinating chromosome segregation during cell divisions (Allshire and Karpen 2008). In most species, the rDNA are also embedded in heterochromatin (McStay 2016) and in Drosophila species, the rDNA locus is generally located in the X pericentromere (Stage and Eickbush 2007). Variation in rDNA copy number is associated with reduced translation capacity in *D. melanogaster* (Mohan and Ritossa 1970; Terracol and Prud'homme 1986). Pericentric heterochromatin may also contain piRNA clusters—discrete loci rich in

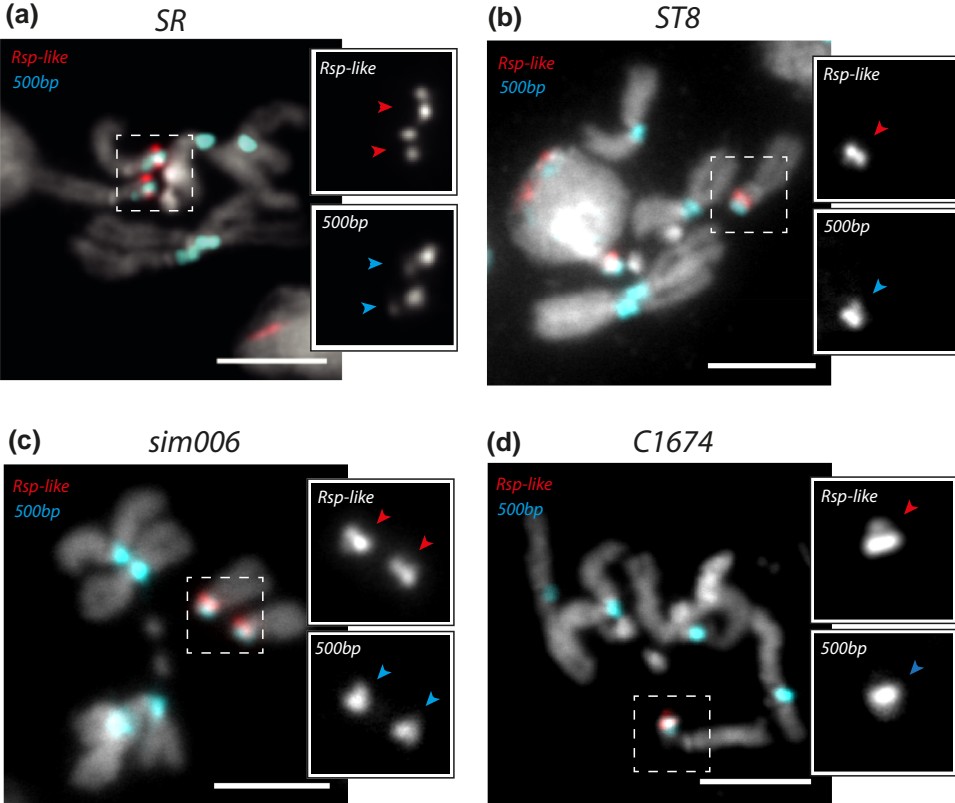

**Fig. 3.** FISH on mitotic chromosomes from larval brains of (a) SR, (b) ST8, (c) sim006, and (d) C167.4 strains. We used oligopaint probes targeting the *Rsp-like* (first inset row) and *500-bp* (second inset row) satellites. The scale bar represents 5 μm. The inset zooms in on the X chromosome with a single focus of *Rsp-like* and *500-bp* in each strain. The arrows within the insets point to each focus associated with the X chromosome.

fragments of transposable elements and other repeats that generate precursors for the small RNAs that are important for the silencing of transposable element activity all over the genome (Brennecke et al. 2007; Aravin et al. 2008). Complex satellite DNAs like those involved in these rearrangements also generate piRNAs that may play a role in establishing heterochromatin in the early embryo (Wei et al. 2021). Finally, while gene density in heterochromatin is generally low, species like *D. melanogaster* do contain hundreds of protein-coding genes in these regions (Marsano et al. 2019), some of which are essential (Devlin et al. 1990; Gatti and Pimpinelli 1992). For some of these genes, a heterochromatic environment is essential for their proper expression and structural rearrangements can disrupt their function (Wakimoto and Hearn 1990; Eberl et al. 1993) and the function of nearby euchromatic genes (Elgin and Reuter 2013).

Structural variation in pericentric heterochromatin can also have global effects on genome stability and regulation. Large blocks of heterochromatin can act as a sink for heterochromatin proteins, titrating them away from other genomic locations (Tartof et al. 1984; Dimitri and Pisano 1989; Eissenberg et al. 1990; Wallrath and Elgin 1995; Francisco and Lemos 2014; Brown et al. 2020). One potential consequence of this sink effect is through its impact on the transcription of euchromatin genes and transposable elements, both which may ultimately impact individual fitness (Francisco and Lemos 2014; Abramov et al. 2016; Nguyen and Bachtrog 2021; Huang et al. 2022).

On the one hand, our observation is concerning. Having different variants of the pericentric heterochromatin segregating in a single isolate might introduce both genetic and phenotypic variation to experiments. It also raises the question of the reproducibility of the results between laboratories. It is important to keep track of, and report, the origin of each isolate. Because the variation we described here is not easy to assay and thus difficult to control for, we recommend limiting potential variation within isolates by periodically re-isogenizing strains. We caution researchers to consider the impact this structural variation may have on their experiments.

On the other hand, this is an intriguing observation. While we expect structural rearrangements in heterochromatic sequences within and between species, these X pericentromeres we study here are highly dynamic even within a single isolates of inbred *D. simulans* strains. Our observations raise several questions. Why is this region particularly unstable? Is this instability specific to the X pericentromere? Is it specific to *D. simulans*? Further investigation will be necessary to better understand the dynamics of structural variation in pericentric heterochromatin and its consequences. The structural rearrangements we describe here are likely associated with genome instability and may represent a unique opportunity to better understand factors promoting the disruption of heterochromatin structure in general. The mechanisms involved in generating these structural rearrangements may be similar to those associated with structural variations involved in human diseases.

## Data availability

Strains are available upon request. The authors affirm that all data necessary for confirming the conclusions of the article are present within the article, figures, and tables.

Supplemental material available at GENETICS online.

## Acknowledgments

We are grateful to Colin Meiklejohn, Daven Presgraves, Peter Andolfatto, and Catherine Montchamp-Moreau for generously sharing fly stocks and to Grace Lee and members of the Larracuente lab for discussion.

## Funding

National Science Foundation grant number MCB-1844693.

## Conflicts of interest

The author(s) declare no conflict of interest.

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
