## [Peer Review File · Genetics]

High levels of intra-strain structural variation in *Drosophila simulans* X pericentric heterochromatin

Cécile Courret and Amanda Larracunte

NOTE: The reviews and decision letters are unedited and appear as submitted by the reviewers.

In extremely rare instances and as determined by a Senior Editor or the EIC, portions of a review may be redacted. If a review is signed, the reviewer has agreed to no longer remain anonymous.

The review history appears in chronological order.

Review Timeline:

Submission Date:	2023-06-29
Editorial Decision:	2023-07-25
Resubmission Received:	2023-08-12
Accepted:	2023-09-14

July 24, 2023

GENETICS-2023-306308

High levels of intra-strain structural variation in *Drosophila simulans* X pericentric heterochromatin

Dear Dr. Courret:

Two experts in the field have reviewed your manuscript, and I have read it as well. While each reviewer had some positive comments, they raised a number of major concerns that make the manuscript unacceptable for publication in its current form. In particular, Reviewer 1 asks whether the observation of polymorphisms could result from drift/selection in small-population lab stocks that could occur by well known mechanisms (e.g. unequal crossovers between repeats of the heterochromatic blocks, where the observed difference between white stocks and wild-types could simply be coincidental.) Reviewer 2 raises additional questions about the interpretation of the FISH data, in particular whether the quality of the staining provides sufficient resolution to score repeat numbers based on the images presented in the manuscript, and also questions whether the observations would be of broad interest. You can read their reviews at the end of this email.

If you believe that you can address the comments of the reviewers, we would consider a substantially revised manuscript. Alternatively, if you wish, I could explore the possibility of transferring your manuscript to G3. The requirements for publishing in G3 would be dependent on consultation with editors from that journal, which would be based on the current reviews. Please let me know if you would like to choose that route.

If you do choose to resubmit a revised manuscript, please include:

1. A clean version of your manuscript;
2. A marked version of your manuscript in which you highlight significant revisions carried out in response to the major points raised by the editor/reviewers (track changes is acceptable if preferred);
3. A detailed response to the editor's/reviewers' feedback and to the concerns listed above. Please reference line numbers in this response to aid the editor and reviewers.

Please note that paper would likely be sent back out for review.

Additionally, please ensure that your resubmission is formatted for GENETICS
<https://academic.oup.com/genetics/pages/general-instructions>

Follow this link to submit the revised manuscript: Link Not Available

Sincerely,

Jack Bateman
Associate Editor
GENETICS

Approved by:
Jeff Sekelsky
Senior Editor
GENETICS

Reviewer #1 (Comments for the Authors (Required)):

This is a review of "High levels of intra-strain structural variation in *Drosophila simulans* X pericentric heterochromatin," submitted by Courret and Larracuenta (manuscript # 2023-306308).

The manuscript describes characterization of polymorphisms - detected by FISH - in X-chromosome-linked centric heterochromatin. These polymorphisms comprise copy-number variations of the satellites 500-bp and Rsp-like repeats and, in this case, copy-number variations refer to the number of discrete blocks of repeat (rather than copy number of individual repeats in arrays, which were not characterized in this work). The authors detect polymorphisms on chromosomes within inbred laboratory strains, even those of different isolates of the same characterized alleles (of white), for which approximate dates of separation of strains can be determined.

The data, specifically the image quality, is high and adequate conclusions can be drawn, although at the size presented in the

manuscript it may be too-low for readers of hard-copies of the research report; it may be helpful to alter the figure style to better demonstrate the data.

The authors should explain their evidence, or justification, to exclude drift (or selection), which would be expected to be pronounced in laboratory stocks which are usually held at low population numbers, are oftentimes recovered from few individuals, are occasionally refreshed from stock collections or sub-isolates within the lab, etc. I am concerned that the vagaries of stock maintenance in laboratories may produce differences over time that can be interpreted as rapid generation of polymorphisms. This is my major concern with the work: what justifies the interpretation that these polymorphisms are truly *de novo*? I agree that it may not be feasible to isogenize X chromosomes and observe new polymorphisms (although it may be), but because a simpler alternative explanation is not excluded by the observations here, the interpretation that these polymorphisms are recent and recurrent is not solely justified (e.g., Lines 120-121 and 160-162).

Relevant to this point, something about the biologies of the 500-bp and Rsp-like satellites seems in order. For Rsp in *Drosophila*, anyway, there is strong selection in the presence of even weak SD systems, which has been provisionally linked to functioning of the piRNA processing pathway. A discussion of the context of this biology would strengthen the interpretive aspect of this work.

I do not understand how the population estimates from Table I were determined. It appears that these are allele frequencies, but simple statement of that would clarify the situation for me. From Table I, are the determinations consistent between all mitotic spreads of each individual? That would seem to be a necessity for population allele frequencies. But the statement in lines 147-149 suggests otherwise. I'm simply not sure if that sentence is purely hypothetical or there is some evidence to support this conclusion.

Other requests/recommendations:

It would be helpful, especially for non-*Drosophila* researchers, to convert years of time since separation of strains to approximate generations. This can be assumed to be about 3-4 weeks, but different laboratories may have different protocols for strain transfer, temperature of maintenance, etc. This would allow all readers to more-easily contextualize the authors' conclusions.

I recommend adherence to standard genetics nomenclature, specifically genetic elements (e.g., rDNA, chromosome names like X and Y) should be italicized.

It would be helpful to include a brief history of strains SR, ST8, sim006, and C167.4. I understand that these are described in another paper (which is cited), but for ease a brief description of their genotype, the date of wild-isolation, and (critically) the size of the founder population/isolation is warranted. Again, back to the alternative hypothesis that the observed polymorphisms of the white alleles are due to drift, if these other (non-white) strains were isogenized or from pairwise matings, the authors' observation that these four "wild" strains do not exhibit any such polymorphisms would be the expectation.

Table I should include the number of spreads, in addition to the number of individuals from which those spreads were obtained. The assumption is that only females were assessed, but that should be stated and, if not true, how male-versus-female spreads were treated in determining the frequencies reported in this table.

I would encourage addition of references to Andy Clark's recent work on satellite polymorphisms. I would also recommend a better description of the source-sink model for PEV and heterochromatin. The cited papers are not the best. Henikoff 1996 points out flaws in the original papers published by Tartof (1988) and Eissenberg (1989) and Dmitri (1989), which themselves offer more-complex ideas. Perhaps this is beyond the scope of the paper the authors wish to write but, at a minimum, the earlier papers should also be cited as they are the origin of the ideas.

Finally, I do not see the value that Supplemental Figure 1 brings. It addresses a question - the detection of polymorphisms - only obliquely. It would be better to do a simple quantification of Rsp-like or 500-bp sequences from the Illumina sequencing. I frankly doubt that there would be any meaningful data there, either, but it would be a pointed and directed analysis of the point the authors mean to make.

-Keith A. Maggert

Reviewer #2 (Comments for the Authors (Required)):

Please see attached document.

Summary:

Courret and Larracuent investigate the structural arrangement of pericentric heterochromatin (specifically, Rsp-like and 500-bp X-linked satellites) in laboratory *D. simulans* strains. The authors first investigate two white- laboratory strains using DNA FISH on larval neuroblasts and find evidence for high heterogeneity, not only between laboratory strains, but also between individuals from a single strain. They investigate wild-type laboratory strains and do not find the same evidence, indicating that this structural rearrangement might be limited to those with white gene mutations. This is an intriguing short study that fittingly raises more questions than it answers. Given that it may be a phenomenon of certain mutant laboratory strains of *D. simulans*, and that there is no evidence yet that it is a more widespread phenomenon (in other genotypes or species or even chromosomes), it's not clear that this information will be of broad utility/interest. In addition, the microscopy data presented could be due to staining artifacts, especially because the neuroblast spreads are not clear. I suggest that at minimum the authors confirm the observed rearrangements using a different tissue in the same strains (e.g. salivary gland polytene chromosomes).

Strengths:

- The manuscript is well-written, clear, concise, and easy to read
 - o The paragraph from 142-153 seems out of place and belongs in the introduction.
- I applaud the authors for presenting negative results and suggesting reasonable conclusions, based on the data.
- There is an inconsistency/typo between text and figure legend: line 112 claims that strains w501-AML and -PA are both polymorphic for 2/3 foci, while the figure legend (Fig 1) claims 1/3 foci for -AML.

Major concerns:

- A major concern is the quality of staining and the risk for false conclusions. The DAPI imaging is very faint, so sometimes it is difficult to determine if foci are on one or several chromosomes. I am also concerned about staining artifacts. For example, Figure 2B #2—the authors suggest a 1/3 focus heterozygote, but Rsp-like could be 2/3 and 500bp could be 1/1. How are the authors making this call?
- Why did the authors choose neuroblast spreads for cytogenetic analyses over, say salivary gland polytene chromosomes, which might give better resolution and the same structural information? If the authors investigated salivary gland as a parallel tissue in the same strains, it would provide either important confirmatory evidence for the neuroblast conclusions OR indicate structural variability between tissues, which is also impactful (and potentially even more interesting).
- Only heterozygous individuals are presented in the figures, but presumably the authors also saw homozygosity. Was this at expected Mendelian ratios within a strain, which

would strengthen the argument for chromosome polymorphisms within a strain? Only the chromosome focus frequencies are given in Table 1.

Minor concerns:

- The authors suggest that the organization of Rsp-like and 500bp satellites is “always similar (i.e., in adjacent blocks)” —lines 104-5, and that we expect colocalization between these two satellites. But how do they know this, especially given the rearrangements claimed in this study? It certainly seems that they give similar signals most of the time, but not always (e.g. Fig 1A #1, or Fig 2B #1). If these satellites look different, which one is “correct” (e.g. Fig 2B #1 – Rsp-like shows two foci, while 500bp shows one)? This is one reason I have concerns about staining and interpretation.
- Given that the investigation focuses on X-linked satellites, do the authors see any sex-specific differences? Are all data from XX females?
- The authors should present the genotyping data (lines 159-160) in the supplement and methods (e.g. primer sequences).
- Table 1 column header should read “proportion of spreads with number of foci” rather than “number of foci,” which is confusing. The table also has a column for “no. of individuals”—is this individual larvae or neuroblast spreads? Do spreads from the same individual give consistent results?
- The authors should explain some history of the “wild type” *D. simulans* strains used in the study (lines 166-7). These strains do not have white mutations, but do carry strain-specific differences. These are cited in the methods text (Courret 2022), but it would be helpful to have relevant genotypic details in the context of this study.

Dear Drs. Bateman and Sekelsky,

Thank you for handling the review of our brief investigation. We appreciated the reviewers' feedback and believe that it helped us greatly improve the manuscript. We respond to each comment point-by-point below in blue italics, and indicate where we made revisions in the manuscript.

Reviewer #1 (Comments for the Authors (Required)):

This is a review of "High levels of intra-strain structural variation in *Drosophila simulans* X pericentric heterochromatin," submitted by Courret and Larracuente (manuscript # 2023-306308).

The manuscript describes characterization of polymorphisms - detected by FISH - in X-chromosome-linked centric heterochromatin. These polymorphisms comprise copy-number variations of the satellites 500-bp and Rsp-like repeats and, in this case, copy-number variations refer to the number of discrete blocks of repeat (rather than copy number of individual repeats in arrays, which were not characterized in this work). The authors detect polymorphisms on chromosomes within inbred laboratory strains, even those of different isolates of the same characterized alleles (of white), for which approximate dates of separation of strains can be determined.

The data, specifically the image quality, is high and adequate conclusions can be drawn, although at the size presented in the manuscript it may be too-low for readers of hard-copies of the research report; it may be helpful to alter the figure style to better demonstrate the data.

Response: Thank you for your feedback. We have enlarged the figures so that they should be easier to interpret in print. Please let us know if you have further suggestions.

The authors should explain their evidence, or justification, to exclude drift (or selection), which would be expected to be pronounced in laboratory stocks which are usually held at low population numbers, are oftentimes recovered from few individuals, are occasionally refreshed from stock collections or sub-isolates within the lab, etc. I am concerned that the vagaries of stock maintenance in laboratories may produce differences over time that can be interpreted as rapid generation of polymorphisms. This is my major concern with the work: what justifies the interpretation that these polymorphisms are truly *de novo*? I agree that it may not be feasible to isogenize X chromosomes and observe new polymorphisms (although it may be), but because a simpler alternative explanation is not excluded by the observations here, the interpretation that these polymorphisms are recent and recurrent is not solely justified (e.g., Lines 120-121 and 160-162).

*Response: Thanks for your comments. Unfortunately, we cannot precisely date these alleles or the rate that this type of structural variation is generated at present. For example, we cannot exclude that these variants were generated in the strain >10 years ago and sorted independently in the different isolates and that the frequencies are affected by selection and drift and that those pressures may vary from lab-to-lab and over time. These are very interesting questions that we would like to explore, but we feel that the experiments required to make those estimations are beyond the scope of this short report. However, we suspect that these are recent *de novo* mutations because of (1) the homozygosity in other parts of the X chromosome, (2) they occur on distinct X chromosomes (with independent*

histories), and (3) because they are multiallelic within strains (e.g., one isolate segregates for 1 and 2-foci, and another segregates for 2- and 3-foci).

We revised the manuscript to make it clear that we cannot date or estimate the rate of these structural mutations but they are cumulative evidence that they are most likely recent.

Here is the additional paragraph in the discussion lines 208-220.

‘We cannot completely rule out the possibility that these variants were already segregating in the original strains and then sorted differently between lab isolates. Isolates of the same strain maintained in different labs can also accumulate isolate-specific TE landscapes over short periods of time (Rahman et al. 2015). Labs may differ in their maintenance conditions, which may impose different selection pressures. Further experiments are necessary to determine the mutation rate in the X pericentromere. We suggest that a recent origin of the structural variants we describe here appear likely, based on multiple observations. First, if there was pre-existing structural variation we would expect more similarity between the w501-AML and w501-DP isolates based on their recent history, than between w501-AML and w501-PA. Second, two independent strains (w^{501} and w^{XD1}) exhibit structural polymorphism in the same region, suggesting that this X pericentric heterochromatin may experience genomic instability. Finally, the two white strains where we see the variation are highly inbred compared to the four non-white strains which do not have detectable structural polymorphisms.’

To further support our hypothesis, we now include an estimate of the per-site heterozygosity for both the white and non-white strains, as a proxy for the isogenization of those strains. Our analyses confirmed that both w^{501} and w^{XD1} are highly inbred strains, compared to the 4 non-white strains.

Here is the corresponding paragraph, lines 187-200 :

‘Isogenization should purge any segregating sequence variants (including structural ones) within strains, although sequence variation may exist due to: 1) mutations that accumulate over time while strains are maintained in labs (Lack et al. 2016); 2.) residual heterozygosity from incomplete inbreeding or linkage to balanced deleterious mutations that cannot be made homozygous. To determine if the structural polymorphism correlates with the extent of inbreeding of each strain, we estimate the per-site heterozygosity (H) of the X chromosome in available genomic data (Hu et al. 2013; Meiklejohn et al. 2018; Courret et al. 2023). Despite being polymorphic in the X pericentromere, we estimate very low levels of per-site heterozygosity across the X chromosome arm in wXD1-CM ($H=1.254 \times 10^{-5}$) and w501-PA ($H=5.93 \times 10^{-5}$). The non-white strains appear less inbred—ST8 ($H=0.000468$), SR ($H=0.000733$) and C167.4 ($H=0.000459$), which are similar to a previous estimate for the sim006 strain ($H=0.00039$) (Kim et al. 2021).

Therefore, the structural polymorphism is in the strains with the lowest heterozygosity across the X chromosome arm, further supporting our hypothesis that the structural variants arose recently and may be associated with genomic instability in the X pericentromere.”

Our main objectives in this short communication are to let the community know that the variation exists in different inbred lines each with extremely low heterozygosity (wXD1-CM $H=1.254 \times 10^{-5}$; w501-PA $H=5.93 \times 10^{-5}$) it is very difficult to see and control for, and that we speculate that not accounting for this type of variation may affect experiments.

Relevant to this point, something about the biologies of the 500-bp and Rsp-like satellites seems in order. For Rsp in *Drosophila*, anyway, there is strong selection in the presence of even weak SD systems, which has been provisionally linked to functioning of the piRNA processing pathway. A discussion of the context of this biology would strengthen the interpretive aspect of this work.

*Response: Thanks for this suggestion. We appreciate the point to bring in more biology, as we suspect that this variation could be consequential. The Rsp-like satellite is not the same as Rsp, which is the target of SD in *D. melanogaster*. We don't know this Rsp-like satellite in *D. simulans* to be involved in some kind of meiotic drive system but it would not surprise us to learn that it too was caught up in some genomic conflict. We added sentences to the discussion about the potential roles for satellite DNAs and their small RNAs generally in establishing heterochromatin in early embryos (lines 244-246) and organizing the chromocenter (lines 233-234).*

I do not understand how the population estimates from Table I were determined. It appears that these are allele frequencies, but simple statement of that would clarify the situation for me. From Table I, are the determinations consistent between all mitotic spreads of each individual? That would seem to be a necessity for population allele frequencies. But the statement in lines 147-149 suggests otherwise. I'm simply not sure if that sentence is purely hypothetical or there is some evidence to support this conclusion.

Response: Thanks for pointing this out. Indeed the number in Table 1 corresponds to allele frequencies within each isolate. The counts are the number of individuals (i.e., brains) that we used for the FISH assay. For each individual brain, all of the spreads we examined were consistent.

We clarified this point in the legend of Table 1, which now reads:

*“Table 1: A summary of structural variation involving the 500-bp and Rsp-like satellites in the X chromosome pericentric heterochromatin within and between isolates of *D. simulans* strains. The isolate identities for w501 and wXD1 are indicated in parentheses. We report the number of individuals (i.e., brains, which includes both males and females) tested: all spreads examined within an individual brain were consistent (see Materials and Methods). We report the proportion of 1-, 2-, or 3-focus X chromosomes among individuals from each isolate. The detailed genotype of each individual tested is presented in SupTable1. “*

We also provide further details in the Methods (line 307-312)

'We analyzed 4-10 mitotic spreads for each individual brain, to determine without ambiguity the number of foci carried by the X chromosomes. We confirmed that all spreads within an individual brain had the same number of foci. To estimate the allele frequency in each isolate, around 20 individual brains were tested, both male and female (full genotype details in SupTable1). The frequency reported in Table 1 corresponds to the frequency of each type of X chromosome among all individual brains tested.'

Lines 147-149 were referring to our attempt to detect these polymorphisms in genomic data where often multiple individuals are pooled for library preparation, so our ability to detect large polymorphic copy number variations (if they exist) especially when relying on read pileup in Illumina data, will depend on their frequency in the pools of individuals used for DNA extraction.

We clarified this point in the manuscript (lines 148-154).

'Structural polymorphisms involving large blocks of the Rsp-like and 500-bp satellite repeats may generally be detectable through differences in read depth (Larracuenta 2014). However, when these polymorphisms exist within a single isolate, they are not obvious in genomic data (Supplemental Figure 1). In our analysis of sequencing libraries created from pooled individuals, detecting alternative alleles based on read depth is extremely challenging, as it will depend on the frequency of alternative alleles in the pools.'

Other requests/recommendations:

It would be helpful, especially for non-Drosophila researchers, to convert years of time since separation of strains to approximate generations. This can be assumed to be about 3-4 weeks, but different laboratories may have different protocols for strain transfer, temperature of maintenance, etc. This would allow all readers to more-easily contextualize the authors' conclusions.

Response: Great idea, we now attempt to estimate a number of generations assuming 13-17 generations/year (based on 3-4-week flip schedule).

I recommend adherence to standard genetics nomenclature, specifically genetic elements (e.g., rDNA, chromosome names like X and Y) should be italicized.

Response: Sorry about that. We think we corrected each occurrence but let us know if we missed any.

It would be helpful to include a brief history of strains SR, ST8, sim006, and C167.4. I understand that these are described in another paper (which is cited), but for ease a brief description of their genotype, the date of wild-isolation, and (critically) the size of the founder population/isolation is warranted. Again, back to the alternative hypothesis that the observed polymorphisms of the white alleles are due to drift, if these other (non-white) strains were isogenized or from pairwise matings, the authors' observation that these four "wild" strains do not exhibit any such polymorphisms would

be

the

expectation.

Response: Good suggestion. We now list the date of collection for each of these strains and that they originated as isofemale lines.

While both white and non-white strains were isogenized, white-eyed lab strains started with a single mutant X chromosome with that white allele and we assumed that there would generally be more starting variation in the wild-type strains compared to the white-eyed strains. As a proxy, we estimate the average per-site heterozygosity across the X chromosome based on the genomic data that we have for two white strains (w501-PA and wXD1-CM) and 3 of the non-white strains (SR, ST8 and C1674). As expected, both white strains have a lower heterozygosity than the non-white strains across their entire X chromosome, despite being polymorphic in their pericentromere. This is consistent with our hypothesis that these structural variations arose de novo recently in these strains. We now include these results in lines 187-200.

“Isogenization should purge any segregating sequence variants (including structural ones) within strains, although sequence variation may exist due to: 1) mutations that accumulate over time while strains are maintained in labs; 2.) residual heterozygosity from incomplete inbreeding or linkage to balanced deleterious mutations that cannot be made homozygous. To determine if the structural polymorphism correlates with the extent of inbreeding of each strain, we estimate the per-site heterozygosity (H) of the X chromosome in available genomic data. Despite being polymorphic in the X pericentromere, we estimate very low levels of per-site heterozygosity across the X chromosome arm in wXD1-CM ($H=1.254 \times 10^{-5}$) and w501-PA ($H=5.93 \times 10^{-5}$). The non-white strains appear less inbred—ST8 ($H=0.000468$), SR ($H=0.000733$) and C167.4 ($H=0.000459$), which are similar to a previous estimate for the sim006 strain ($H=0.00039$; Kim et al. 2021).

Therefore, the structural polymorphism is in the strains with the lowest heterozygosity across the X chromosome arm, further supporting our hypothesis that the structural variants arose recently and may be associated with genomic instability in the X pericentromere.”

Table I should include the number of spreads, in addition to the number of individuals from which those spreads were obtained. The assumption is that only females were assessed, but that should be stated and, if not true, how male-versus-female spreads were treated in determining the frequencies reported in this table.

Response: All spreads from the same individuals had the same number of foci and we only used the number of individual brains to estimate frequencies. We are worried that it might be confusing to include the number of spreads in the table itself (as we only use to assure consistency within brains but not in the frequency estimations), so we instead now indicate in the Materials and Methods that we surveyed 4-10 mitotic spreads per individual brain and that we surveyed both males and females. We based the allele frequency on the total number of X chromosomes (i.e., 1 for male and 2 for female). We provide the number of individuals for each genotype in a new supplemental table S1.

I would encourage addition of references to Andy Clark's recent work on satellite polymorphisms. I would also recommend a better description of the source-sink model for PEV and heterochromatin. The cited papers are not the best. Henikoff 1996 points out flaws in the original papers published by Tartof (1988) and Eissenberg (1989) and Dmitri (1989), which themselves offer more-complex ideas.

Perhaps this is beyond the scope of the paper the authors wish to write but, at a minimum, the earlier papers should also be cited as they are the origin of the ideas.

Response: Thanks very much for these suggestions. We now cite these references.

Finally, I do not see the value that Supplemental Figure 1 brings. It addresses a question - the detection of polymorphisms - only obliquely. It would be better to do a simple quantification of Rsp-like or 500-bp sequences from the Illumina sequencing. I frankly doubt that there would be any meaningful data there, either, but it would be a pointed and directed analysis of the point the authors mean to make.

Response: We struggled with deciding on the best way to represent the (negative) genomic analysis. Copy number analyses can't quite get at what we are after because the estimates based on read depth will be an average of all alleles that went into the library and the 500-bp satellite is present on nearly all chromosomes. The purpose of that figure is to illustrate that it is difficult to deduce the allelic states at the satellites based on genomic analysis.

-Keith A. Maggert

Reviewer #2 (Comments for the Authors (Required)):

Summary:

Courret and Larracuente investigate the structural arrangement of pericentric heterochromatin (specifically, Rsp-like and 500-bp X-linked satellites) in laboratory *D. simulans* strains. The authors first investigate two white- laboratory strains using DNA FISH on larval neuroblasts and find evidence for high heterogeneity, not only between laboratory strains, but also between individuals from a single strain. They investigate wild-type laboratory strains and do not find the same evidence, indicating that this structural rearrangement might be limited to those with white gene mutations. This is an intriguing short study that fittingly raises more questions than it answers. Given that it may be a phenomenon of certain mutant laboratory strains of *D. simulans*, and that there is no evidence yet that it is a more widespread phenomenon (in other genotypes or species or even chromosomes), it's not clear that this information will be of broad utility/interest. In addition, the microscopy data presented could be due to staining artifacts, especially because the neuroblast spreads are not clear. I suggest that at minimum the authors confirm the observed rearrangements using a different tissue in the same strains (e.g. salivary gland polytene chromosomes).

Strengths:

- The manuscript is well-written, clear, concise, and easy to read
- o The paragraph from 142-153 seems out of place and belongs in the introduction.

Response: Thanks for this feedback. This paragraph is based on both our intuition given published literature and our analysis of all genomic data we could get our hands on in an attempt to detect these variations computationally. We elected to put it in the results because our conclusions are based on

our analyses. We now add some more context for this analysis in response to reviewer 1's comments, so we hope that it feels less out of place now.

- I applaud the authors for presenting negative results and suggesting reasonable conclusions, based on the data.
- There is an inconsistency/typo between text and figure legend: line 112 claims that strains w501-AML and -PA are both polymorphic for 2/3 foci, while the figure legend (Fig 1) claims 1/3 foci for -AML.

Response: Thanks very much for pointing this out. We corrected the typo.

Major concerns:

- A major concern is the quality of staining and the risk for false conclusions. The DAPI imaging is very faint, so sometimes it is difficult to determine if foci are on one or several chromosomes. I am also concerned about staining artifacts. For example, Figure 2B #2—the authors suggest a 1/3 focus heterozygote, but Rsp-like could be 2/3 and 500bp could be 1/1. How are the authors making this call?

Response: Thanks for pointing out that you did not find the images straightforward to interpret. We tried to increase the interpretability of all images.

The DAPI was a bit faint, so we increased the brightness uniformly to make it easier to see. We added arrows pointing to each probe signal focus belonging to the X chromosome to avoid ambiguity. We hope this will clarify the interpretation of the images.

Because we base our conclusions on multiple images/observations per individual brain, the spreads (to us) are very clear though. We think we see what your concern is about the number of foci. We treated with a hypotonic solution, which increases the separation of sister chromatids. Sometimes you can see both chromatids separately, which is the case in Figure 2B for Rsp-like, but it is still clear that it is a single allelic focus on the chromosome.

- Why did the authors choose neuroblast spreads for cytogenetic analyses over, say salivary gland polytene chromosomes, which might give better resolution and the same structural information? If the authors investigated salivary gland as a parallel tissue in the same strains, it would provide either important confirmatory evidence for the neuroblast conclusions OR indicate structural variability between tissues, which is also impactful (and potentially even more interesting).

Response: We agree that it would be very interesting to explore whether instability at satellites leads to variation in chromosome structure between tissues. This will be difficult to do with current technologies. Unfortunately, heterochromatin is underreplicated in polytene chromosomes and so these satellites appear in the chromocenter, which has no information about its organization on the chromosome. In order to be interpretable, we need chromosome spreads like mitotic or meiotic chromosomes. Larval neuroblasts have a high mitotic index, so these chromosomes are readily visible and ideal for visualizing structural rearrangements in heterochromatin.

- Only heterozygous individuals are presented in the figures, but presumably the authors also saw homozygosity. Was this at expected Mendelian ratios within a strain, which would strengthen the argument for chromosome polymorphisms within a strain? Only

the chromosome focus frequencies are given in Table 1.

Response: We chose to provide images for heterozygotes only to highlight the different alleles but we did also see homozygotes. We now provide the detailed genotype of each individual in SupTable1.

Minor concerns:

- The authors suggest that the organization of Rsp-like and 500bp satellites is “always similar (i.e., in adjacent blocks)”—lines 104-5, and that we expect colocalization between these two satellites. But how do they know this, especially given the rearrangements claimed in this study? It certainly seems that they give similar signals most of the time, but not always (e.g. Fig 1A #1, or Fig 2B #1). If these satellites look different, which one is “correct” (e.g. Fig 2B #1 – Rsp-like shows two foci, while 500bp shows one)? This is one reason I have concerns about staining and interpretation.

Response: Thanks for this comment. We meant that the two satellites seem to co-vary on the chromosome, so if there are two foci of 500-bp, there are two foci of Rsp-like. This is because they are adjacent on the chromosome. We didn't see any genotypes where 500-bp and Rsp-like did not co-vary in the number of foci. We hope that we now make this clear on lines 109-100.

In Fig1A #1 both Rsp-like and 500-bp show 2 foci. On the 500-bp inset there is indeed a third spot, but this one belongs to an autosome which is just adjacent and not on the X chromosome. We now add arrows that point to the foci that are associated with the X chromosome to avoid ambiguity. We also increase the DAPI signal to make more clear to which chromosome each foci belongs to. We hope this helps with interpretation of our figure.

Similarly, for Fig2B#1, while there do seem to be two foci for Rsp-like, those actually correspond to the two sister chromatids. Analyzing multiple spreads for each individual helps us be confident in our allele assignments for each brain.

- Given that the investigation focuses on X-linked satellites, do the authors see any sexspecific differences? Are all data from XX females?

Response: We only showed heterozygotes females because we wanted to maximize the number of alleles represented in the images, however we did assay males of each genotype and homozygous females as stated above. We did not observe any sex-specific differences. We now provide the detailed genotypes for each individual in SupTable1. Thanks for the prompt.

- The authors should present the genotyping data (lines 159-160) in the supplement and methods (e.g. primer sequences).

Response: We added the primer sequences to the methods in addition to the coordinates –thanks for that request.

- Table 1 column header should read “proportion of spreads with number of foci” rather than “number of foci,” which is confusing. The table also has a column for “no. of individuals”—is this individual larvae or neuroblast spreads? Do spreads from the same individual give consistent results?

Response: Thanks for pointing out confusion with this table. Reviewer 1 also pointed out that it wasn't clear. In Table 1, we report the proportion of X chromosomes (not spreads) carrying each allele. For each brain, 4-10 spreads were analyzed to determine the type of X chromosome carried by the individual. All spreads from the same individual were consistent.

We modified the table and clarified the legend to make it clear that we report the proportion of each type of X chromosome.

- The authors should explain some history of the “wild type” *D. simulans* strains used in the study (lines 166-7). These strains do not have white mutations, but do carry strain-specific differences. These are cited in the methods text (Courret 2022), but it would be helpful to have relevant genotypic details in the context of this study.

Response: This is a great suggestion, also made by reviewer 1. We added information about the location and date of collection for each non-white strain in the Methods (starting line 289):

*“We also used other non-white *D. simulans* isofemale strains: SR (collected from Seychelles in 1981), ST8 (collected from Tunisia in 1983), C167.4 (collected from Kenya in 1973), sim006 (collected from California in 1961) (described in Courret et al. 2023). “*

September 13, 2023
RE: GENETICS-2023-306421

Dr. Cécile Courret
University of Rochester
Biology
River Campus
Hutchison hall
Rochester, New York 14620

Dear Dr. Courret:

Congratulations! We are delighted to inform you that your manuscript entitled "High levels of intra-strain structural variation in *Drosophila simulans* X pericentric heterochromatin" is acceptable for publication in GENETICS. Many thanks for submitting your research to the journal.

To Proceed to Production:

1. Format your article according to GENETICS style, as discussed at <https://academic.oup.com/genetics/pages/general-instructions>, and upload your final files at <https://genetics.msubmit.net>. Staff will review your files and be in touch if there are any questions.
2. After staff check your files, they will be transmitted to OUP for processing. You will then receive an email with a link to sign your license to publish. Please add jnls.author.support@oup.com and genetics.oup@novatechset.com (or the domains @oup.com and novatechset.com) to your email program's "safe senders" list. Publication cannot occur till the license is signed. Invoices are generated after the license is signed.
3. Your manuscript will be published as-is (unedited-as submitted, reviewed, and accepted) at the GENETICS website as an Advanced Access article and deposited into PubMed shortly after receipt of source files and the completed license to publish. Please notify sourcefiles@thegsajournals.org if you do not wish to publish your article via Advanced Access.
4. We invite you to submit an original color figure related to your paper for consideration as cover art. Please email your submission to the editorial office or upload it with your final files. You can submit a small-sized image for evaluation, and if selected, the final image must be a TIFF file 2513px wide by 3263px high (8.375 by 10.875 inches; resolution of 600ppi). Please avoid graphs and small type.
5. Let us know if this paper is from a recently-established lab (within the last 5 years) so it can be considered for extra visibility.

If you have any questions or encounter any problems while uploading your accepted manuscript files, please email the editorial office at sourcefiles@thegsajournals.org.

Sincerely,

Jack Bateman
Associate Editor
GENETICS

Approved by:
Jeff Sekelsky
Senior Editor
GENETICS

note: Please add jnls.author.support@oup.com and genetics.oup@novatechset.com (or the domains @oup.com and novatechset.com) to your email program's "safe senders" list. You will be contacted by both at various points during the production process.

Review comments (if applicable):

Reviewer #2 (Comments for the Authors (Required)):

This is a re-review of Courret and Larracuenta's "High levels of intra-strain structural variation in *Drosophila simulans* X pericentric heterochromatin."

The authors addressed my concerns point by point. In general, my central concern - that the polymorphisms characterized in this work are the result of drift in laboratory strains - remains unaddressed. That possibility would mean that the polymorphisms that are characterized as rapidly-arising and consistent among unrelated genotypes are not consequential, nor do they reveal something fundamental about the underlying etiology of structural variation. However, it is possible that the concern I raise is not itself addressable, nor is it a concern. To wit, the authors argue that the alternative (drift) is unlikely, and I accept their arguments. The authors also make a larger point, which to me stands even if drift is the explanation, that many of the strains we believe to be isogenic are in fact not. That itself is an important point, and the authors make sure to state that clearly. Overall, the manuscript now reads as more "circumspect," presenting the data, the support for the hypothesis, but also includes a balanced interpretation of other possibilities. This is the way science should be presented, and I find this version to be both interesting and supported.

-Keith A. Maggert